# A Purified Aspartic Protease from *Akkermansia Muciniphila* Plays an Important Role in Degrading Muc2

**DOI:** 10.3390/ijms21010072

**Published:** 2019-12-20

**Authors:** Xin Meng, Wencheng Wang, Tianqi Lan, Wanxin Yang, Dahai Yu, Xuexun Fang, Hao Wu

**Affiliations:** 1Key Laboratory for Molecular Enzymology and Engineering of Ministry of Education, College of Life Science, Jilin University, 2699 Qianjin Street, Changchun 130012, China; mengxin17@mails.jlu.edu.cn (X.M.); w1148893960@163.com (W.W.); SkyQiii@163.com (T.L.); yinglang6918@126.com (W.Y.); yudahai@jlu.edu.cn (D.Y.); 2Vascular Biology Program, Department of Surgery, Boston Children’s Hospital and Harvard Medical School, Boston, MA 02115, USA

**Keywords:** Amuc_1434, *Akkermansia muciniphila*, aspartic protease, mucin–degradation, localization

## Abstract

*Akkermansia muciniphila* can produce various mucin-degrading proteins. However, the functional characteristics of these proteins and their role in mucin degradation are unclear. Of the predicted protein-coding genes, *Amuc_1434*, which encodes for a hypothetical protein, is the focus in this study. A recombinant enzyme Amuc_1434 containing the 6× His-tag produced in *Escherichia coli* (hereinafter termed Amuc_1434*) was isolated to homogeneity and biochemically characterised. Results showed that the enzyme can hydrolyse hemoglobin with an activity of 17.21 U/μg. The optimal pH and temperature for hemoglobin hydrolysis of Amuc_1434* were found to be around 8.0 and 40 °C, respectively. Amuc_1434* is identified as a member of the aspartic protease family through the action of inhibitor pepstatin A. Amuc_1434* promotes the adhesion of colon cancer cell line LS174T, which can highly express Muc2. Significantly Amuc_1434* can degrade Muc2 of colon cancer cells. Amuc_1434 is mainly located in the colon of BALB/c mice. These results suggest that the presence of Amuc_1434 from *Akkermansia muciniphila* may be correlated with the restoration of gut barrier function by decreasing mucus layer thickness.

## 1. Introduction

Mucins are a family of high molecular weight glycosylated proteins that protects epithelial cells and exists on multiple organ surfaces [1,2,3]. Mucins can participate in cell signal transduction [4], and function in cell adhesion, immune recognition, metastasis and has been correlated with the prognosis of epithelial tumors [5]. Thus, the expression of mucins affect cell differentiation and tumorigenesis. Currently, twenty mucins have been characterized, and they are clustered into two types [6]: Secretory mucinous proteins and transmembrane mucinous proteins.

Mucin 2 (Muc2) belongs to the type of secretory mucinous proteins. It is normally expressed only in the colon and rectum but not in normal ovarian epithelial cells, and generally considered as an intestinal-associated antigen [7]. Muc2 is also known as the intestinal mucin, which is associated with intestinal goblet cells, consisting of a variable number of vertically repeated amino acid sequences (VNTR) [8]. The structure of secretory mucin gene is highly consistent with human vWF factor, and its function shares similarity with vWF factor [9]. When cells become cancerous under pathological conditions, this type of mucin may be involved in the migration and diffusion of tumor cells by combining with matrix collagen components [8,10]. Muc2 is the main component of the protective layer of colon mucosa [11], which can inhibit intestinal inflammatory response, and thus prevents the development of intestinal tumors [12,13]. It is involved in the occurrence and development of colorectal cancer (CRC), which ranks the third cause of mortality of cancers in the world [14,15]. The expression of Muc2 is reduced in the common colorectal non–mucinous adenocarcinoma [16,17], but it is highly expressed and has a high degree of malignancy and invasiveness in the colorectal mucinous adenocarcinoma [18,19,20]. Muc2 protein is a major component of pseudomyxoma peritonei (PMP) mucinous tumor tissue [21]. High Muc2 is positively correlated with breast [22,23,24], bladder [25], gastric [26], and ovarian cancers [27]. Therefore, Muc2 degradation may be an effective treatment for some cancers.

Reduction of Muc2 is mainly caused by proteolytic enzymes [28,29]. Currently, an increasing number of studies suggest that bacterial enzymes greatly contribute to the luminal proteases [30,31]. *Akkermansia muciniphila* has been reported as a new generation of probiotics [32]. It is well adapted to the mucous layer and accounts for 1–5% of fecal microorganisms in healthy adults [33,34]. Therefore, *Akkermansia muciniphila* plays a key role in maintaining gastrointestinal balance and intestinal barrier integrity. Previous studies have shown that *Akkermansia muciniphila* abundance is negatively correlated with some metabolic disorders in humans and mice [35,36]. For example, inflammatory bowel disease [37], obesity [38], autism [39], and type 2 diabetes [40], etc have all been implicated. *Akkermansia muciniphila* can use mucus as the sole source of carbon and nitrogen by producing several mucolytic enzymes [33,41]. Therefore, *Akkermansia muciniphila* also plays an important role in mucin degradation [42]. However, the mechanisms how *Akkermansia muciniphila* degrades mucins are still poorly understood. This is due to the limited number of functional characteristic proteins and few studies have linked the specific characteristics of these proteins to ability of bacterial strains to degrade and utilize mucins. The genome of *Akkermansia muciniphila* ATCC BAA–835 has 2176 predicted protein-coding sequences [43]. Of the predicted protein-coding genes, 61 known proteins [43,44] (3%) can be clustered into four categories: Glycosyl hydrolases, sulfatases, proteases, and sialidases); and 242 hypothetical proteins (11%) may be involved in the degradation and treatment of mucins [43]. Among them, the functional properties of these putative mucinolytic proteins and their role in mucin degradation have not yet been reported.

The hypothetical protein-coding gene *Amuc_1434* from *Akkermansia muciniphila* was targeted based on the sequence analysis conducted in present study. We analyzed the basic enzymatic properties of Amuc_1434* and found that it belongs to aspartic protease family, the activity of Amuc_1434* was 17.21 U/μg when hemoglobin was used as the substrate. The optimal pH and temperature were 8.0 and 40 °C, respectively. We also studied the relationship between Amuc_1434* with Muc2. The association between Amuc_1434* and Muc2 was detected by the adhesion of Amuc_1434* to LS174T cells, which was positively correlated with the concentration of Amuc_1434*. Amuc_1434* can degrade Muc2 by in vitro experiments, including Western blot, enzyme-linked immunosorbent assay (ELISA), and immunofluorescence imaging. We also explored the localization of Amuc_1434 in the intestinal tract of BALB/c mice. Amuc_1434 was found to be primarily located in the colon of BALB/c mice.

This study preliminarily explores the degradation mechanism of Muc2 by *Akkermansia muciniphila*. We speculate that the presence of Amuc_1434 secreted by *Akkermansia muciniphila* may play an important role in the intestine and may be correlated with the restoration of gut barrier function by modulating mucus layer thickness.

## 2. Results

### 2.1. Purification of Amuc_1434*

The putative protein Amuc_1434* with aspartic protease conserved motifs Asp-Thr-Gly (DTG) and Leu-Leu_Gly (LLG) was selected from the *Akkermansia muciniphila* genome database, following amplified from genomic DNA of *Akkermansia muciniphila*, and cloned successfully into an *Escherichia coli* expression vector. Firstly, the soluble expression of Amuc_1434* (the amino acid sequence is shown in Figure 1A) was successfully obtained. As expected, a 6xHis-tag fused to the N-terminus of Amuc_1434* could generate Amuc_1434*, and nearly all expressed proteins were soluble. His–tagged Amuc_1434* was further purified using optimizing step gradients of imidazole concentration, and identified by sodium dodecyl sulfate polyacrylamide gel electrophoresis (SDS–PAGE). The observed molecular weight (MW) of the purified protein (~50 kDa) was in accordance with the theoretical mass as shown in Figure 1B. The purified protein was further verified by Western blot using a primary rat antibody against 6×His tag as shown in Figure 1C.

### 2.2. Amuc_1434* Activity Test and Kinetic Study

Given that the sequence of Amuc_1434* contains the conserved “DTG” and “LLG” motifs of the aspartic protease family [45], it is perhaps not suprising that Amuc_1434* has the aspartic protease activity. Thus, we measured its protease activity using hemoglobin as the substrate. Hydrolysis of bovine hemoglobin assay showed that the purified protein Amuc_1434* could hydrolyze hemoglobin with an activity of 17.21 U/μg.

Next, we tested the kinetic assay of Amuc_1434*. The catalytic activity of Amuc_1434* was determined at 37 °C and pH 8.0 with different hemoglobin concentrations ranging from 0.015 to 0.6 μmol/mL. Amuc_1434* exhibited Michaelis–Menten kinetics in hemoglobin hydrolysis (Figure 2A). The results were plotted in accordance with Lineweaver–Burk (Figure 2B). The enzyme worked with a good correlation coefficient square (R^2^) of 0.9986, Michaelis–Menten–type kinetics with K_m_ = 0.22 μM, V_max_ = 0.26 μM/min, and K_cat_ = 0.18 min^−1^. The catalytic efficiency (K_cat_/K_m_) of the enzyme was found to be 0.82/min/μM.

### 2.3. Effect of pH on the Activity of Amuc_1434*

The activity of purified Amuc_1434* was studied over a pH range of 4.0–10.0 with the substrate as hemoglobin at 37 °C (Figure 3). The purified Amuc_1434* showed relatively high activity in the range of pH 6.0–8.0 with maximum activity at pH 8.0. The enzyme activity of Amuc_1434* reduced with decreasing pH over pH 4.0–8.0. Significantly, it decreased at pH 8.6 and lost almost entire activity at pH 10.0.

### 2.4. Effect of Temperature on the Activity and Half-Life of Amuc_1434*

The effect of temperature on Amuc_1434* activity was determined by performing activity assays at pH 8.0, using 0.3 μmol/mL of hemoglobin as substrate in the range of 30–90 °C (Figure 4A). The enzyme was active over a temperature range of 40–50 °C with an optimal activity at 40 °C, and completely inactive at above 70 °C. The thermal stability of the enzyme was at 40 °C, with a half–life of more than 4 h (Figure 4B).

### 2.5. Effects of Inhibitors on Amuc_1434*

To study the properties of the purified Amuc_1434*, several common protease family inhibitors were selected to investigate whether the hydrolysis hemoglobin activity of Amuc_1434* can be blocked by these inhibitors. The inhibitors included pepstatin A (aspartate protease inhibitor, 0.02 mM), ethylene diamine tetraacetic acid (EDTA, chelating agent, metalloprotease inhibitor, 10.0 mM), E–64 (cysteine protease inhibitor, 0.01 mM), and phenylmethylsulphonyl fluoride (PMSF; serine protease inhibitor, 1.0 mM).

As shown in Table 1, the addition of EDTA and PMSF could improve Amuc_1434* activity. Conversely, its activity was slightly inhibited by E–64, and greatly inhibited by pepstatin A, suggesting that Amuc_1434* could have the aspartic protease activity.

Considering that the sequence of Amuc_1434* contains the conserved “DTG” and “LLG” motifs of the aspartic protease family [45], We would like to investigate if Amuc_1434* could be completely inhibited by Pepstatin A. Strikingly, Amuc_1434* completely lost its activity when pepstatin A concentration was higher than 0.04 mM, indicating that Amuc_1434* belongs to the aspartic protease family (Figure 5). It was reported that the weak inhibition of aspartic protease by E-64 was probably due to a small non-specific effect from the carrier solvents and/or the high concentration of the inhibitors themselves [46].

### 2.6. Association between Amuc_1434* and Muc2

Association between Amuc_1434* and Muc2 was studied by using a cell adhesion assay. First, we identified the expression of Muc2 in the two cell lines by Western blot, and found that LS174T expressed Muc2, while HeLa barely expressed Muc2 (Figure 6A). Next, a cell adhesion assay was performed to identify the compatibility and specificity of Amuc_1434* to Muc2. A gradient of concentrations of Amuc_1434* were added in the 96-well plates following seeded equal amounts of LS174T cells into the same plates, and then incubated overnight at 4 °C. Amuc_1434* showed a dose-dependent attachment to LS174T cells. Interestingly, the number of adhesion LS174T cells with Amuc_1434* was significantly increased when Amuc_1434* concentrations were ≥4 μg/mL (Figure 6B). However, the adhesion of HeLa cells was not detected to be correlated with Amuc_1434* (Figure 6B). These data suggest that the LS174T cells expressing Muc2 could be attached by Amuc_1434*.

### 2.7. Amuc_1434* Degradation of Muc2

Amuc_1434* was derived from the myxotropic bacterium *Akkermansia muciniphila*. Thus, we investigated the ability of Amuc_1434* to degrade mucins. We also employed LS174T cell line, which secretes a large amount of Muc2, to study Muc2 degradation ability by Amuc_1434*. Untreated and Amuc_1434*–treated Muc2 were subjected to agarose gel electrophoresis, transferred onto a nitrocellulose membrane and probed with Muc2 antibody. Muc2 was degraded by Amuc_1434* in a concentration-dependent manner by Western blot (Figure 7A). Significantly, degradation of Muc2 band with Amuc_1434* treatment is much higher than that of Muc2 band without Amuc_1434* treatment. Thus, the higher the Amuc_1434* concentration, the stronger the degradation ability. The results of ELISA also showed a similar result. The remaining Muc2 content detected by the ELISA kit decreased with increasing Amuc_1434* concentration (Figure 7B). When Amuc_1434* (32 μg/mL) was co–incubated with Muc2 at 37 °C for 3 h, 42% Muc2 was degraded. We conducted immunofluorescence experiments to study the ability of the Amuc_1434* protease to degrade Muc2. The cell membrane was stained green with 3,3′-dioctadecyloxacarbocyanine perchlorate (DIO). After incubation with PE secondary antibody for 1 h, the Muc2 of LS174T cells incubated with the Muc2 antibody was marked in red. The number of extracellular Muc2 in Amuc_1434* treated cells was significantly reduced compared to untreated cells as shown in Figure 7C.

### 2.8. Amuc_1434 Localization

Immunohistochemistry was used to study the localization of Amuc_1434 in the intestinal tract of mice. In the immunohistochemical experiment, the Amuc_1434 (brown–yellow colour) was observed in the rectum, cecum, duodenum and colon of BALB/c mice (Figure 8A). However, the colon of BALB/c mice showed significantly higher of Amuc_1434 concentration than theses in the rectum, cecum and duodenum. The average optical (AO) density value was calculated as follows: AO = IOD (Integrated Optical Density)/AREA. Higher AO corresponds to higher positive expression. As shown from the statistical figure (Figure 8B), the AO value calculated from the colon section was the largest, the AO values calculated from the caecum and duodenum sections were basically the same, and the AO value calculated from the rectum section was the smallest. These data were significantly confirmed that Amuc_1434 is mainly located in the colon in mice.

## 3. Discussion

Muc2 is also known as the intestinal mucin, which is associated with intestinal goblet cells [8]. It is normally expressed only in the colon and rectum [7] and is the main component of the protective layer of colon mucosa [11]. Mucus hypersecretion was linked to bacterial overgrowth and the induction of inflammatory responses that could promote tumor development [47,48]. Muc2 overexpression could pass the production of mucous barrier, protect tumor cells (such as colon cancer) from recognition by anti–tumor immune effectors and contribute to a malignant phenotype [49]. Therefore, the expression of Muc2 was a valuable evaluative indicator for colon cancer–related diseases. *Akkermansia muciniphila* is a probiotics at the degradation of human intestinal mucin [36]. It contains more than 300 genes for putative proteins involved in mucin degradation, but the nature and function of most of these proteins are unknown [43]. In the present study, we are the first to report on the production and purification of a hypothetical protein Amuc_1434* secreted by *Akkermansia muciniphila*, and to determine its biochemical properties and ability to degrade Muc2.

The target protein was purified by isopropyl β-d-Thiogalactoside (IPTG) induction, purification and desalination. The purified protein was found to be homogeneous with a molecular mass of 50 kDa (Figure 1B,C). Firstly, we characterized the protease properties of Amuc_1434*. As Amuc_1434* contains the conserved “DTG” and “LLG” sequence motifs of the aspartic protease family [45], we measured its protease activity using hemoglobin as the substrate. We found that this enzyme could efficiently hydrolyze hemoglobin. In partial acidic buffer (NaAc–HAc) and alkaline buffer (Glycine–NaOH), the enzymatic activity was significantly lower than in the neutral and slight alkaline buffer (PBS and Tris-HCl). The enzyme exhibits high activity in the weak alkaline buffer, such as pH 7.5 PBS and 8.0 Tris-HCl. The optimal protease activity was observed at pH 8.0 (Figure 3). Although most aspartic proteases are commonly found to have optimal activities in the acidic pH, we repeatedly obtained the result. We also used di-sodium hydrogen phosphate (Na_2_HPO_4_)-citric acid buffer (pH 2.2–8.0), and Amuc_1434* again showed its optimal activities at pH 8.0 (data not shown). The optimal temperature and half-life studies showed that it exhibited maximal activity at 40 °C and it could keep active after 4 h at optimum temperature (Figure 4A,B). The effects of various inhibitors on enzyme activity were studied. PMSF, EDTA and E–64 are the classic serine, metallo, and cysteine protease inhibitors, respectively, to characterize purified enzymes. In the presence of these three inhibitors, the enzyme still had or retained most of its activity (Table 1). However, the activity of Amuc_1434* was greatly decreased and even totally inactivated when performing this with aspartic protease inhibitor pepstatin A (Figure 5), suggesting Amuc_1434* could belong to aspartic protease family.

Classical aspartic proteinase includes the active site consensus sequence motifs DTG/DSG. Despite a relatively low amino acid sequence identity with other aspartic proteinases, the Amuc_1434* sequence contains the typical hallmarks of aspartic proteinases. This includes the active site consensus sequence motifs (DTG) and the one hydrophobic-hydrophobic-Gly motif of the Psi-loops, indicating that Amuc_1434* displays the similarity with RC1339/APRc from *Rickettsia conorii*, a novel aspartic protease with properties of retropepsin-like enzymes [45]. Given the importance of degradation of Muc2 by Amuc_1434* secreted from *Akkermansia muciniphila*, the exactly degradation mechanisms, the potential pathophysiology and clinical relevance of this degradation warrant further investigation.

Next, we explored the relationship between Amuc_1434* and Muc2, from the point of view that Amuc_1434 originated from *Akkermansia muciniphila* mucophilin. In cell-adhesion experiments, we found that Amuc_1434* promoted the adhesion of LS174T cells with high expression of Muc2. The higher the concentration of Amuc_1434*, the more adherent cells can be quantified at OD_570 nm_. On the contrary, Amuc_1434* did not promote HeLa cells adhesion with low expression of Muc2. Therefore, we concluded that there was an association between Amuc_1434* and Muc2, and that Muc2 was an influencing factor of Amuc_1434* in favor of LS174T cells adhesion, leading to the increase of OD_570 nm_ (Figure 6A,B). we also studied the effect of Amuc_1434* from *Akkermansia muciniphila* on the expression of Muc2. Western blot experiments confirmed that Amuc_1434* could degrade Muc2 in a concentration–dependent manner (Figure 7A). ELISA was performed to detect quantitatively the ability of Amuc_1434* to degrade Muc2 (Figure 7B). The residual Muc2 content decreased with increasing Amuc_1434* concentration after co–incubation. The results of immunofluorescence assay showed that the addition of Amuc_1434* reduced the Muc2 content on the membrane surface (Figure 7C). The distribution of Amuc_1434 in mice tissues was detected by immunohistochemistry to explore the physiological function of Amuc_1434. The results showed a strong Amuc_1434 positive immunoreaction in the colon of BALB/c mice (Figure 8A,B), suggesting Amuc_1434 was mainly located in the colon.

To the best of our knowledge, the present study is the first to find a Muc2 degradation protein from *Akkermansia muciniphila*. We found that Amuc_1434* could associate with Muc2 and participate in its degradation. We also found that Amuc_1434 was mainly distributed in the colon of BALB/c mice, but what role it plays in the colon is still unknown.

In order to better understand the functions of Amuc_1434, the following problems need to be explored in the future: whether Amuc_1434, as a mucin-degrading enzyme, can affect the growth of *Akkermansia muciniphila*? Considering that Muc2 is closely associated with tumours, can Amuc_1434 control the occurrence and development of tumours by degrading Muc2? If so, which pathways are involved in its regulation?

The results in this study suggest that the presence of Amuc_1434 from *Akkermansia muciniphila* may be correlated with the restoration of gut barrier function by decreasing mucus layer thickness.

## 4. Materials and Methods

### 4.1. Materials

LS174T colon cancer cell lines were obtained from the American Type Culture Collection (Manassas, VA, USA). Dulbecco’s modified eagle’s medium (MA0212, DMEM, including 4.5 g/L D-Glucose, 584 mg/L L-Glutamine and 110 mg/L Sodium Pyruvate) and fetal bovine serum (A31607) for LS174T cell culture were purchased from Dalian Meilun Biotechnology Co. LTD. (Dalian, China) and Life Technologies (Scotland, UK), respectively. BCA (bicinchonininc acid) Protein Assay Kit (C503051) was purchased from Sangon Biotech Co. Ltd. (Shanghai, China). Human Muc2 ELISA kit (GE–S54876) was purchased from Bestgene (Changchun, China). Primary antibody (HC201-01, anti–β-actin mouse polyclonal IgG antibody) and the secondary antibody (HS201-01, HRP (horseradish peroxidase)-conjugated goat anti–mouse antibody) for cell attachment assay were purchased form Transgen Biotech Co. LTD. (Beijing, China). Primary antibody (801101, anti–Amuc_1434 rabbit polyclonal IgG antibody) and the secondary antibody (S0001, HRP-conjugated goat anti–rabbit antibody) for immunohistochemistry were purchased from Affinity Biosciences(Queensland, Australia).

### 4.2. Induction and Expression of Proteins

The plasmid of Amuc_1434* was entire gene synthesized by BGI Genomics (Shenzhen, China) Co. Ltd. to achieve the successful expression of recombinant protein Amuc_1434*. Pet25b–His (6× histidine-tagged expression vector) was selected as the vector, and BamHI, XhoI as the restriction sites. The successfully constructed plasmids were transferred to *Escherichia coli* competent cells (*E.coli* rosetta, DE3) (Novagen, Beijing, China) [50]. *E.coli* rosetta (DE3) as an expression host was transformed and cultured in the LB medium (100 μg/mL chloramphenicol or 100 μg/mL ampicillin) at 37 °C overnight with shaking. Four milliliters overnight cultures were added to 400 mL LB medium containing 100 μg/mL chloramphenicol and 100 μg/mL ampicillin and grown with shaking of 180 rpm at 37 °C until OD = 0.6, then cultures were added to 1 mM IPTG to induce protein expression with shaking at 37 °C for 4 h. Then, the cells were collected by centrifugation at 1520 g for 20 min at 4 °C and resuspended in PBS. The cells were treated with mild sonication on ice. The lysates were divided into two parts by centrifugation at 13,680 g for 10 min at 4 °C. Then, the supernatant was transferred to the fresh tube (soluble protein samples) and stored at 4 °C. The precipitation was dissolved in 10 mL of solution buffer (20 mM Tris–HCl at pH 8.0, 8 M Urea, 500 mM NaCl) for 2–3 h at room temperature, and then centrifuged at 13,680 g for 10 min. The supernatant was collected to a fresh tube and then stored at 4 °C.

### 4.3. Purification and Identification of Proteins

Ni–Sepharose column was selected for protein purification. After 2 mL resin wrapped Ni–Sepharose column was pre–balanced with the equilibrium buffer (0.2 M Na_2_HPO_4_, 0.2 M NaH_2_PO_4_ at pH 7.5), the unpurified soluble protein samples were placed on the column. The purified protein was eluted with a sequential imidazole series (20, 50 and 80 mM) after the unpurified soluble protein samples were cycled 1 h. Then boiled in protein-loading buffer for 10 min at 100 °C, separated by 12% reducing sodium dodecyl sulfate–polyacrylamide gel electrophoresis (SDS–PAGE), and dyed with Coomassie brilliant blue. The remaining purified protein was added with 2% glycerin and stored at −80 °C.

### 4.4. Western Blot

The successful purification of protein was applied to Western blot. Firstly, it was separated by 12% SDS–PAGE. Secondly, the proteins in the gel were blotted onto polyvinylidene difluoride (PVDF) membranes by electrophoretic transfer. The electrophoretic transfer buffer included Tris (3 g/L) and glycine (14.4 g/L). Thirdly, the blot membrane was blocked in 10% skimmed milk with shaking for 1 h. Then incubation with primary mouse antibody against 6× His tag (ab18184, Abcam (Cambridge, UK), Dilution ratio 1:1000, using TBS (25 mM Tris, 150 mM NaCl)) was performed for the night at 4 °C. After washing, the PVDF membranes were incubated for 1 h with HRP-conjugated goat anti–mouse secondary antibody (ab205719, Abcam, Cambridge, UK) at RT and then washed three times with TBST (25 mM Tris, 150 mM NaCl, 0.05% Tween-20). The protein was detected with the super sensitive electrochemiluminescence (ECL) luminescence reagent (MA0186, Meilunbio, Dalian, China).

### 4.5. Activity Test of Purified Amuc_1434*

The protein concentration was also measured by BCA Protein Assay Kit [51] (C503051, Sangon Biotech (Shanghai, China) Co. Ltd.) after the imidazole was removed by ultrafiltration. The enzymatic activity of Amuc_1434* (70 μg/mL) was tested using haemoglobin as the substrate by ultraviolet and visible spectrophotometry. In the assay, 980 μL of PBS buffer, 5 μL of protease and 15 μL of haemoglobin (0.3 μmol/mL) were used. Then the ultraviolet absorbance was recorded at the wavelength of 275 nm. The control had no substrate in the reaction system. The amount of enzyme was required to convert 1 micromole of hemoglobin in 1 min under certain conditions as an activity unit.

### 4.6. Kinetics of Amuc_1434* Catalyzed Reactions

To determine the kinetic parameters, the Amuc_1434* in the catalytic activity was measured under different concentrations of its substrate. The Michaelis constant (K_m_) and the maximum rate (V_max_) of the enzyme–catalyzed reaction were assayed via a double reciprocal (Lineweavere Burk) plot [52] of Amuc_1434* activity and substrate concentrations (0.015–0.6 μmol/mL) at pH 8.0 at 37 °C.
(1)$$\frac{1}{V}=\frac{V_{m}}{V_{\max}}\times \frac{1}{\left[S\right]}+\frac{1}{V_{\max}}$$
where V, V_max_, K_m_, and [S] are reaction rate, maximum reaction rate, Michaelis–Menten constant and substrate concentration in equation (1), respectively. The V_max_ and K_m_ were obtained by the y-intercept (1/V_max_), the x-intercept (−1/K_m_) from a plot of 1/V versus 1/[S] (Lineweaver–Burk plot). K_cat_ is defined to equal V_max_/E, where E is the concentration of enzyme, which was added to the assay.

### 4.7. Effect of pH and Temperature on Amuc_1434* Activity

The kinetic changes of the substrate within 1 min at 37 °C were determined using different buffer systems to identify the optimal pH of the purified Amuc_1434*. The buffer system was as follows: NaAc–HAc (50 mM, pH 4.0–5.5), PBS (50 mM, pH 6.0–8.0), Tris–HCl (50 mM, pH 7.5–8.9) and Glycine–NaOH (50 mM, pH 8.6–10.0). The assay method for measuring activity was as described above. To study the optimum temperature of the purified Amuc_1434*, the Amuc_1434* activity was measured in 50 mM PBS (pH 8.0) at different temperatures (30–90 °C) under the conditions as described above. Experiments of each pH and temperature were repeated for at least three times.

### 4.8. Half-Life Study

In order to determine the half-life of purified Amuc_1434*, Amuc_1434* was incubated at the optimal temperature at different time points. Samples were taken out at different time intervals, and the Amuc_1434* activity was measured using the above method.

### 4.9. Effect of Inhibitors on Proteolytic Activity

The Amuc_1434* and inhibitors (E–64, PMSF, EDTA, pepstatin A) were incubated together at 37 °C for 30 min and the remaining activity of the Amuc_1434* were determined. The Amuc_1434* without inhibitor treatment was used as control, and the relative activity was defined as 100%. The effect of different pepstatin A concentrations on the haemoglobin hydrolysis ability of Amuc_1434* was determined using the above method.

### 4.10. Cell Attachment Assay

The expression of Muc2 in LS174T and HeLa cells line were detected by Western blot with anti–Muc2 rabbit polyclonal IgG antibody (DF8390, Affinity Biosciences), and β-actin as the internal reference. The primary antibody were anti–β-actin mouse polyclonal IgG antibody (HC201-01, Transgen Biotech Co. LTD., Beijing, China), and the secondary antibody was HRP-conjugated goat anti–mouse antibody (HS201-01, Transgen Biotech Co. LTD., Beijing, China). The protein Amuc_1434* of indicated concentrations (from 0 to 16 μg/mL) was dissolved in coating buffer (PBS) and immobilised on a 96-well plate by incubating overnight at 4 °C [53]. After the plates were sealed with 2% bovine serum albumin (BSA) (dissolved in PBS) at 37 °C for 1 h, the LS174T and HeLa cells (4 × 10^4^) were inoculated into the plate, and then incubated for 2 h at carbon dioxide cell incubator. Then the plates were cleaned with PBS, the adherent cells were fixed with 4% paraformaldehyde solution (AR-0211, Beijing Dingguo Changsheng Biotechnology Co. Ltd., Beijing, China) for 30 min at room temperature (RT). After washing the cells, the cells were stained with 0.25% crystal violet (dissolved in PBS) for 10 min. After washing off the excess dye, the cells were observed and photographed under a white light microscope. These cells in each hole were dissolved with dimethyl sulfoxide (DMSO) (100 μL/well) and quantified by spectrophotometry at 570 nm.

### 4.11. Muc2 Degradation Experiment

The similar method described by Szabady et al. [54] was used to detect the activity of Amuc_1434* on cell–related Muc2. LS174T cells were cultured in a cell culture dish for 24 h, the culture solution was discarded, and the monolayer cells were washed and dissolved using 150 μL lysis buffer radio immunoprecipitation assay (RIPA) lysate (P0013B, Beyotime, Beijing, China), PMSF, 1 mM), then left on ice for 30 min. The cells were centrifuged at 13,680× *g* for 5 min at 4 °C. Each 50 μL of supernatant was separately added to five EP (eppendorf) tubes with different volumes (0, 50, 100, 200 and 400 μL) of 50 μg/mL purified Amuc_1434*, and PBS was added to 500 μL of total volume in each sample EP tube, mixed well and then incubated for 3 h at 37 °C. The samples were then separated on gradient (4–15%) PAGE. Western blot was performed in accordance with the above experimental methods to detect the degradation of Muc2, but with the use of different antibodys. Primary antibody were anti–Muc2 rabbit polyclonal IgG antibody (DF8390, Affinity Biosciences), and the secondary antibody were HRP-conjugated goat anti–rabbit antibody (S0001, Affinity Biosciences, Queensland, Australia). Similarly, β-actin was chosen as the internal reference protein.

### 4.12. Sandwich Enzyme-Linked Immunosorbent Assay (ELISA)

Human Muc2 ELISA kit (GE–S54876) was selected to quantify the ability of Amuc_1434* to degrade Muc2 [55,56]. Sandwich ELISA immunoassay. Firstly, the cell suspension was diluted with PBS, and repeated freezing and thawing caused cell damage and released intracellular components. After centrifugation (380 g) for 20 min, the collected supernatant was incubated for 1.5 h at 37 °C with a protease of different final concentrations (from 0 to 32 μg/mL). Each sample hole on the enzyme label-coated plate (mouse anti–human Muc2 monoclonal antibody was used as capture antibody) had added to it 40 μL of sample diluent and 10 μL of the sample to be tested. Then, the sample was added with 100 μL of enzyme label reagent (rabbit anti–human polyclonal antibody (HRP) was used as detection antibody), incubated at room temperature for 1 h, cleaned and then coloured. Exactly 50 μL of chromogenic agent A (0.04 g TMB dissolved in 4 mL DMSO) and B (9.7 mL of citric acid, 10.3 mL of disodium hydrogen phosphate) was added to each well, away from light at 37 °C for 15 min. Then, 50 μL of stop solution was added to stop the reaction. The control well does not include the sample and the enzyme label reagent, and the remaining steps are the same as above. The absorbance of each well was measured at 450 nm wavelength with zero adjustment of control well.

### 4.13. Muc2 Immunofluorescence

Muc2 degradation of LS174T cell surface by Amuc_1434* was detected by immunofluorescence staining [31,57]. LS174T cells (10^6^ cells/mL) were cultured on 6-well tissue culture plates at 37 °C overnight. The supernatant was abandoned and replaced with 1 mL of DMEM containing purified proteases at a final concentration of 30 μg/mL. The control did not contain any enzymes. The cell monolayers were treated for 3 h at 37 °C and 5% CO2. Then, the cells were fixed with 4% paraformaldehyde solution (AR-0211, Beijing Dingguo Changsheng Biotechnology Co. Ltd., Beijing, China) for 30 min, and infiltrated with Triton X–100 (0.01%, diluted with PBS) for 30 min. The cells were washed and sealed with BSA (10%, dissolve in PBS) at room temperature for 30 min. The cells were then incubated for 2 h with a 1:500 dilution of an anti–Muc2 rabbit polyclonal IgG antibody (DF8390, Affinity Biosciences) at 37 °C. The cells were washed three times with PBS and then incubated for 60 min at room temperature with a 1:1000 dilution of PE (phycoerythrin) goat anti–rabbit secondary antibody (in red) (HS121-01, TransGen, China). After further washing with PBS, the membranes were counterstained with a 1:10,000 dilution of DIO (in green) for 10minat room temperature. The excess dye was washed three times with PBS and photographed with a fluorescent microscope. Images obtained from a single channel overlap to produce a composite image.

### 4.14. Immunohistochemistry

Amuc_1434 was expressed in normal mice. To accurately locate Amuc_1434 in the intestinal tract of mice, we isolated the colon, cecum, rectum and duodenum of normal BALB/c mice from the body and prepared sections. In immunohistochemistry [58,59], the tissue sections were incubated overnight at 4 °C with primary antibody (anti–Amuc_1434 rabbit polyclonal IgG antibody) (801101, Affinity Biosciences). The second antibody (HRP-conjugated goat anti–rabbit antibody) (S0001, Affinity Biosciences) was then used, and the sections were lightly counterstained with haematoxylin and eosin. The Amuc_1434 colour was brown–yellow. The nuclei were blue. Capture Optical images were captured with a Zeiss Axioscope microscope fitted with an Axiocam HRm camera. Analysis was performed using the average optical density value of immunohistochemistry with Image–pro plus 6.0 (Media Cybernetics, Inc., Rockville, MD, USA). Each section in each group was randomly selected for at least three fields (×200) for photography. The entire field of view was filled with the organisation, and the background light was the same for each photo. All care and handling of animals were performed in accordance with the guidelines of the Animal Ethics Committee of Jilin University (Approval No. JLUSWLL003, Jilin, China).

### 4.15. Statistical Analysis

Using Graphpad Prism 7.0 statistics. The values are expressed as mean ± standard deviation. Univariate analysis of variance was used for group comparison. * Represent *p*-value < 0.05, ** Represent *p*-value < 0.01, *** Represent *p*-value < 0.001. *p*-value < 0.05 was considered statistically significant.

## Figures and Tables

**Figure 1 ijms-21-00072-f001:**
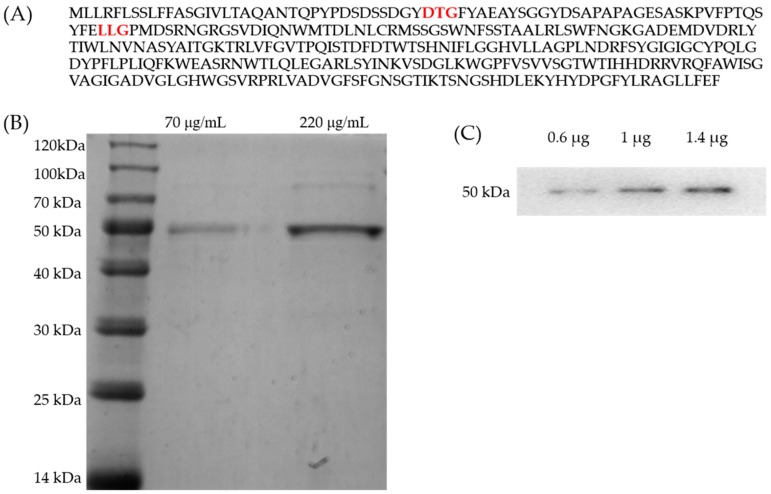
Amuc_1434* amino acid sequences, expression and purification. (**A**) Amuc_1434* amino acid sequence. Red letters represented the conserved motifs of the aspartic protease family. (**B**) Sodium dodecyl sulfate polyacrylamide gel electrophoresis (SDS–PAGE) analysis of the purification of protein Amuc_1434*. An aliquot of the protein after purification (70 μg/mL, left) and following a about 3-fold concentration (220 μg/mL, right) were loaded. The protein purity was higher than 95%, calculated with ImageJ software. (**C**) Western blot analysis of purified Amuc_1434*, loading protein were 0.6, 1 and 1.4 μg, respectively.

**Figure 2 ijms-21-00072-f002:**
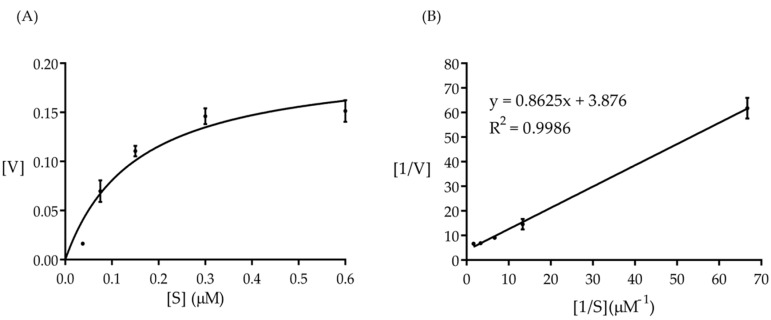
Kinetic analysis of Amuc_1434* hydrolyzed hemoglobin (substrate). (**A**) Michaelis–Menten kinetics plot. The initial reaction rates versus different hemoglobin concentrations. [V] represented reaction rate, [S] represented haemoglobin concentrations. Graphpad Prism software, version 7, was used to fit the data into the Michaelis–Menten equation with a non-linear regression method. (**B**) Double reciprocal (Lineweaver–Burk) plot of the initial rate of hemoglobin hydrolysis by Amuc_1434* versus hemoglobin concentration.

**Figure 3 ijms-21-00072-f003:**
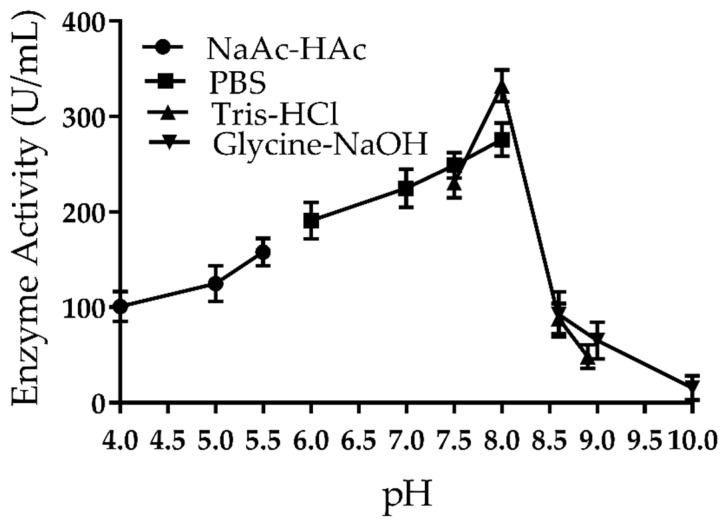
Effect of pH on proteolytic activity of the purified Amuc_1434*. Buffer solutions used for this activity include the following: NaAc–HAc (50 mM, pH 4.0–5.5), phosphate-buffered saline (PBS, 50 mM, pH 6.0–8.0), Tris–HCl (50 mM, pH 7.5–8.9), Glycine–NaOH (50 mM, pH 8.6–10.0). The samples were done in triplicate and the error bars represented the standard deviation of the data.

**Figure 4 ijms-21-00072-f004:**
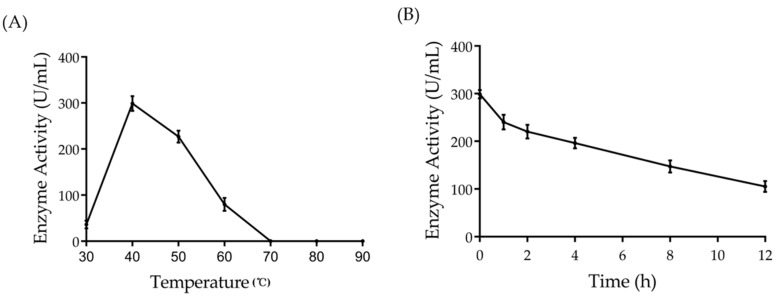
The thermal stability and the half-life of Amuc_1434*. Effect of (**A**) temperature in the range of (range of 30–90 °C) on hydrolyzed hemoglobin activity and (**B**) half-life study of the purified Amuc_1434* at 40 °C. Triplicate samples were measured and error bars represent standard deviation of the data.

**Figure 5 ijms-21-00072-f005:**
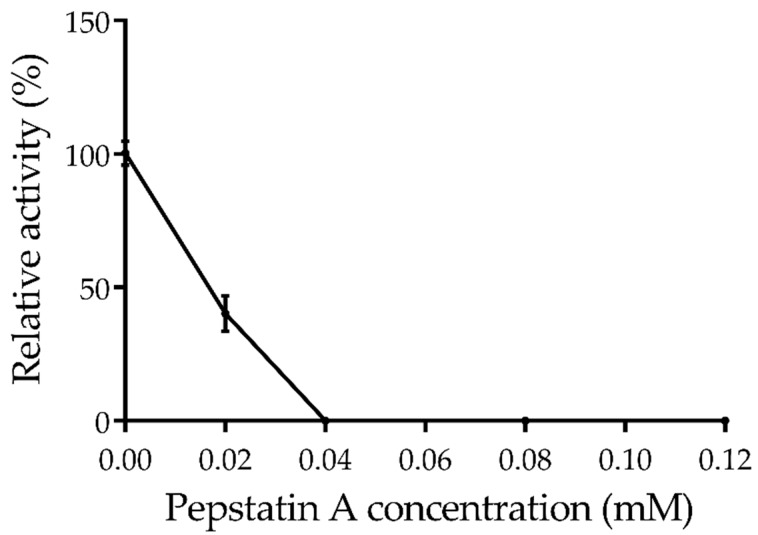
Effect of inhibitors pepstatin A with different concentrations (from 0 to 0.12 mM) on Amuc_1434* activity. Triplicate samples were measured and the error bars represented standard deviation of the data.

**Figure 6 ijms-21-00072-f006:**
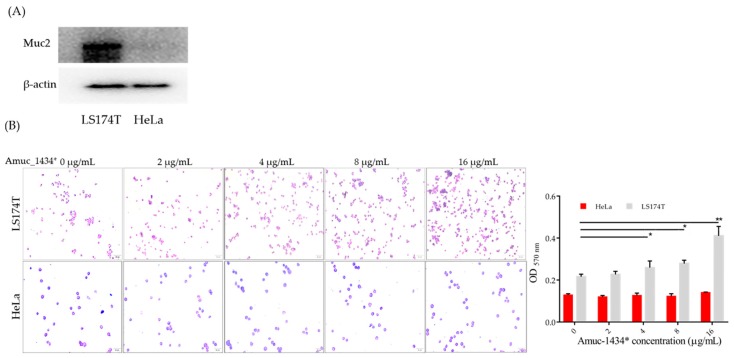
Association between protein Amuc_1434* and Muc2. (**A**) Determined Muc2 expression in LS174T and HeLa cells using Western blot. (**B**) Effect of Amuc_1434* concentration in the range of 0–16 μg/mL on adhesion of LS174T cells. HeLa cells were used as negative control. LS174T and HeLa cells bound to the wells were photographed. Crystal violet was extracted, and the remaining adherent cells were quantified by spectrophotometric absorbance at 570 nm. Scale bar = 50 μm (*n* = 3, *: *p* < 0.05; **: *p* < 0.01). The black line indicated which two groups were compared for significant differences in the column chart.

**Figure 7 ijms-21-00072-f007:**
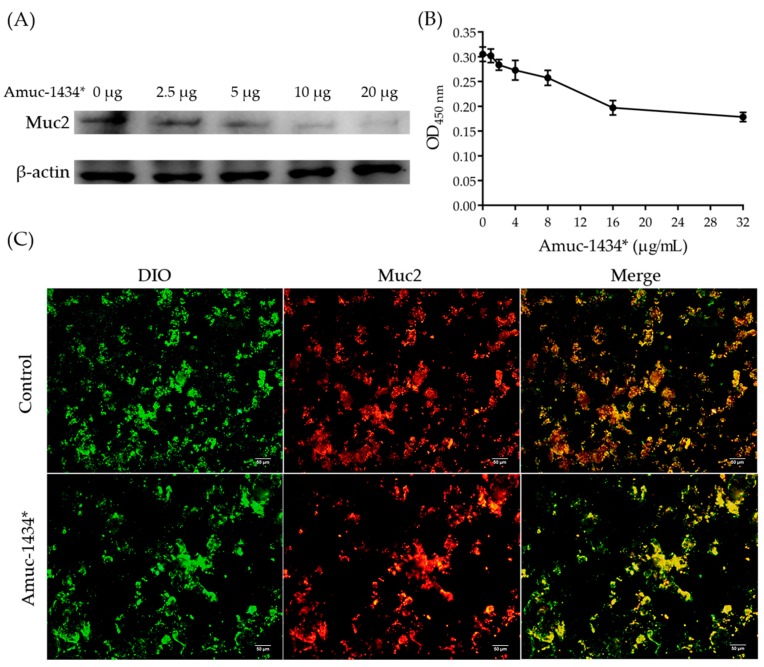
Amuc_1434* degradation of Muc2. (**A**) Western blot analysis of degradation of Muc2 by Amuc_1434* at different Amuc_1434* concentrations (loading protein were 0, 2.5, 5, 10 and 20 μg, respectively). (**B**) Enzyme-linked immunosorbent assay (ELISA) analysis of remaining Muc2 after Amuc_1434* degradation at 450 nm. Triplicate samples were measured and error bars represented standard deviation of the data. (**C**) Immunofluorescence analysis of the Muc2 degradation ability of Amuc_1434*. The membrane was shown in green with 3,3′-dioctadecyloxacarbocyanine perchlorate (DIO). Muc2 was labeled in red. Scale bar = 50 μm.

**Figure 8 ijms-21-00072-f008:**
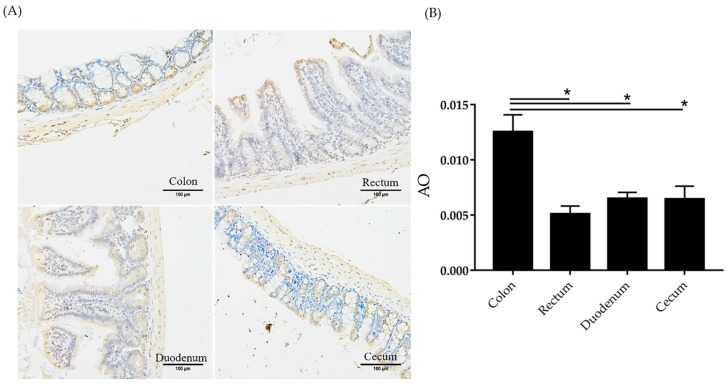
Amuc_1434 localization. (**A**) Immunohistochemical analysis of protein Amuc_1434 colonization in mice intestinal tract. Amuc_1434 showed brown–yellow color, the nucleus was blue. Claybank distribution was greater in the colon than in the other tissues (rectum, duodenum and cecum) of BALB/c mice. (**B**) Expression level of protein in intestinal components by image–pro Plus 6.0 software, average optical (AO) density = IOD (Integrated Optical Density)/AREA. (*n* = 3, *: *p* < 0.05). The black line indicates which two groups are compared for significant differences.

**Table 1 ijms-21-00072-t001:** Effects of inhibitors on protease Amuc_1434* activity.

Compound	Concentration (mM)	Relative Activity (%)
None	-	100.0 ± 9.1
EDTA	10.0	149.9 ± 14.4
PMSF	1.0	126.9 ± 2.6
Pepstatin A	0.02	40.15 ± 6.7
E-64	0.01	88.5 ± 4.7

“-”indicates no inhibitor treatment. The activity of Amuc_1434* was expressed as relative activity, and the activity of Amuc_1434* without inhibitors treatment was defined as 100%.

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
