# Peer review of "A Purified Aspartic Protease from Akkermansia Muciniphila Plays an Important Role in Degrading Muc2"

_ijms, 2019, doi:10.3390/ijms21010072_

Round 1

Reviewer 1 Report

I found this work interesting. . The authors describe the discovery of a new mucin-degrading enzyme with potential application for the study of the microbiota-mucin interaction. This research work spans from the kinetic characterisation to the localisation in the intestine of mice. Despite the interest of the research work, I have found some inconsistences that the authors should explain:

In section 2.2, authors define kinetic activity of the enzyme. In example, authors show a Km in mM. However, the haemoglobin concentration in Figure 2 is expressed in mg/mL, according to the text. The authors should explain how the Vmax, Kcat are obtained from Figure 2. In section 2.3, authors are using different buffers to analyse the activity of the enzyme. They stated that buffer NaAC-HAc could be used in the range of pH 4.0-5.5, however, there is a value for this buffer at pH 6.0 in Figure 3. In addition, Tris-HCl and Glycine-NaOH buffer have a range of pH broader than plotted in the Figure 3. It would be interesting if authors analyse the activity of the enzyme at pH 7.5 in Tris-HCl and at pH 8.6 in Glycine-NaOH. As different enzymatic activities are seen at same pH with different buffers, it could help to detect a buffer-related enzyme activity The figure caption of the Figure 4 is not understood. Regarding to the section B) in this figure caption, what do the authors mean with “stability”? In section 2.5. Authors stated that E-64 “barely inhibit […] Amuc_1434* activity”. According to the Table 1, it reduces the activity of the enzyme to 88.5 %, at 0.01 mM. From the Figure 5, it could be analysed that pepstatin A at 0.01 mM reduces the activity to 72 % approximately. Why the authors discarded the option to check the effect of increasing the concentration of the E64? In section 2.6 and on, authors show results about the SDS-PAGE of β-actin, however, authors did not mention β-actin in the manuscript at all. Even in the methods section. Authors should explain how they isolated the protein and the reasons why they are using it. In Figure 7, authors show immunofluorescence analyses of cell cultures. Why do it appear a yellow colour in the immunoassay of Muc2 (in the middle column)? Also, authors indicate in the figure caption that there is a scale bar for the figure, however I could not find it. In figure 8 (B), the notation of the significant differences do not follows the same format as in figure 6. In the discussion section, authors should discuss the effect of different solvents on the enzymatic activity of the enzyme under analysis. The effect of different inhibitors, the effect that the presence of mucus in the LS174T cells can have in the increase of the absorbance at 570 nm, and so on. In the materials and methods section: Authors should explain which the formulation of the culture cell medium is. The data indicated is not enough. Did the media contain non-essential aminoacids or antibiotics? Which content of glutamine? And so on. The centrifugation speeds should be expressed as “g”, not as rpm, unless the authors provide the radius of the rotor of the centrifuge that they are using As the authors explain in the manuscript that some analyses have been done with purified enzyme, they should explain, how they store the enzyme after the purification step. In assays involving antibodies, authors should check and cite in a proper way which antibodies are they using. In western blot assay, they stated that they used a “primary rat antibodies against […]” and a “HRP goat anti-mouse secondary antibody”. According to that, the secondary antibody used won’t recognize the primary antibody. As the authors have some bands, I presume that it have been a typographic error. In section 4.11, authors stated “western blot was performed in accordance with the above experimental methods”. In that section, “the primary antibodies were anti-Muc2 rabbit” but no secondary antibody is indicated and the secondary indicated in the previous section won’t recognize either the primary antibody cited. In section 4.12, it is not indicated which is the primary antibody, nor the secondary. In that case, it just says “HRP secondary antibody”. In section 4.13 and 4.14, information about antibodies is missing. Authors should give any explanation to the fact that they use PBS to study the enzymatic activity of the enzyme at pH 8.0 instead using Tris HCl, in which, in fact, showed the highest enzymatic activity. In the study of the effect of the inhibitors on the enzymatic activity, authors should explain how they achieve to stop the reaction before measuring the enzymatic activity. In cell attachment assay, authors should indicate which is the composition of the “coating buffer”. Authors should indicate in which solvent is prepared the BSA for sealing the plates, and which solution was used for washing the plates. Also the temperature for the fixation and the solvent in which paraformaldehyde was dissolved. In section 4.12, line 360, authors indicate that they centrifuged the sample but the speed is not indicated. In 4.13 section, authors use CFU to refer to number of cells. Cells do not form colonies. CFU is a microbiological term. Also, please check the solutions and solvents used. In section 4.14, line 396, what does it means (x200)? It means that authors selected three fields with at least 200 cells? The standard streptavidin-peroxidase method implies the presence of a primary biotinylated-antibody or not? As you are using a primary antibody, was it biotinylated? Along the document, authors use some abbreviations not described, in example: DIO, PE, IPTG, ECL, EP, HRP In figure 3, authors should check the spelling of the legend Check the spell of HeLa cells, “L” it is a capital letter.

Reviewer 2 Report

The authors in this paper provide the experimental data on the important role of Amuc_1434, mucin–degrading protein, in degrading
Muc2. Importantly, the results clearly showed that Amuc_1434 promotes the adhesion of colon cancer cell line LS174T, which is a highly expressed Muc2 cell line, and effectively degraded Muc2.

The manuscript to me is, in general, clearly written. The science and technical execution of the study is of good quality. The study is solid and the data, in general, support the conclusions. The theory, logic, and experimental design are easy to follow and in general, make sense.

Minor comments

line 72: change this to present.

Line 78: change in vitro to be italic.

Some figures (fig 3) are missing the significance mark.

Fig 6A: replace the WB image of Muc 2, it is not clear.

Some image: scale bar is missing, show it on the image

Overall, I believe the improved version of the paper will be of interest to the field of colon cancer, Therefore, it should be recommended for publication in IJMS after minor revision

Author Response

Response to Reviewer 2 Comments

The authors in this paper provide the experimental data on the important role of Amuc_1434, mucin–degrading protein, in degrading Muc2. Importantly, the results clearly showed that Amuc_1434 promotes the adhesion of colon cancer cell line LS174T, which is a highly expressed Muc2 cell line, and effectively degraded Muc2.

The manuscript to me is, in general, clearly written. The science and technical execution of the study is of good quality. The study is solid and the data, in general, support the conclusions. The theory, logic, and experimental design are easy to follow and in general, make sense.

Minor comments

Point 1: line 72: change this to present. 

Response 1: Thanks very much for the reviewer’s suggestion. “this” has been revised to “present” (Page 2, Line 72).

Point 2: Line 78: change in vitro to be italic.

Response 2: “in vitro” has been changed to be italic (Page 2, Line 78).

Point 3: Some figures (fig 3) are missing the significance mark.

Response 3: The missing significance mark in Figure 3 has been added accordingly (Page 4, Line 127).

Point 4: Fig 6A: replace the WB image of Muc 2, it is not clear.

Response 4: The old WB image has been replaced by a new and clear WB image of Muc 2 as shown in Fig 6A (Page 6, Line 173).

Point 5: Some image: scale bar is missing, show it on the image

Response 5: The missing scale bar has been added accordingly (Page 7, Line 197, Fig 7C and Page 8, Line 215, Fig 8A).

Point 6: Overall, I believe the improved version of the paper will be of interest to the field of colon cancer, Therefore, it should be recommended for publication in IJMS after minor revision.

Response 6: Thanks very much for the reviewer. Hope the revised version of the paper could be published in IJMS.

Round 2

Reviewer 1 Report

Dear authors,

Many thanks for accepting my suggestions on your work. I think that the manuscript have been greatly improved. However, I will comment just a few things:

In the figure 3, the symbols are not appreciated. Perhaps, you could enlarge the symbols. Also in this figure it is written "glysine" instead of "glicine" in the legend. Could you rewrite the legend?  In the line 393, it says "rabbit anti-human polyclonal antibody" should it say "rabbit anti-mouse [...]" since the primary antibody was "mouse anti-human Muc2 monoclonal antibody"? Could you also check the line spacing along the manuscript? it seems that it has a different line spacing from line 386 to 399 than the line spacing along the document
